# Detection of SARS-CoV-2 Infection in Gargle, Spit, and Sputum Specimens

Eero Poukka,[a] Henna Mäkelä,[a] Lotta Hagberg,[b] Thuan Vo,[a,c] Hanna Nohynek,[a] Niina Ikonen,[b] Kirsi Liitsola,[b] Otto Helve,[a] Carita Savolainen-Kopra,[b] Timothée Dub[a]

[a]Infectious Disease Control and Vaccinations Unit, Department of Health Security, Finnish Institute for Health and Welfare, Helsinki, Finland
[b]Expert Microbiology Unit, Department of Health Security, Finnish Institute for Health and Welfare, Helsinki, Finland
[c]Health Sciences Unit, Faculty of Social Sciences, University of Tampere, Tampere, Finland

Eero Poukka and Henna Mäkelä contributed equally to this article. Author order was determined by a coin toss.

**ABSTRACT** The gold standard for severe acute respiratory syndrome coronavirus 2 (SARS-CoV-2) infection diagnosis is reverse transcription (RT)-PCR from a nasopharyngeal swab specimen (NPS). Its collection involves close contact between patients and health care workers, requiring a significant amount of workforce and putting them at risk of infection. We evaluated self-collection of alternative specimens and compared their sensitivity and cycle threshold ($C_T$) values to those of NPS. We visited acute coronavirus disease 2019 (COVID-19) outpatients to collect concomitant NPS and gargle specimens and had patients self-collect gargle and either sputum or spit specimens the next morning. We included 40 patients and collected 40 concomitant NPS and gargle specimens, as well as 40 gargle, 22 spit, and 16 sputum specimens the next day (2 patients could not produce sputum). All specimens were as sensitive as NPS. Gargle specimens had a sensitivity of 0.97 (95% confidence interval [CI], 0.92 to 1.00), whether collected concomitantly with NPS or the next morning. Next-morning spit and sputum specimens showed sensitivities of 1.00 (95% CI, 1.00 to 1.00) and 0.94 (95% CI, 0.87 to 1.00]), respectively. The gargle specimens had significantly higher mean $C_T$ values of 29.89 (standard deviation [SD], 4.63; $P < 0.001$) and 29.25 (SD, 3.99; $P < 0.001$) when collected concomitantly and the next morning, respectively, compared to NPS (22.07 [SD, 4.63]). $C_T$ values obtained with spit (23.51 [SD, 4.57]; $P = 0.11$) and sputum (25.82 [SD, 9.21]; $P = 0.28$) specimens were close to those of NPS. All alternative specimen collection methods were as sensitive as NPS, but spit collection appeared more promising, with a low $C_T$ value and ease of collection. Our findings warrant further investigation.

**IMPORTANCE** Control of the COVID-19 pandemic relies heavily on a test-trace-isolate strategy. The most commonly used specimen for diagnosis of SARS-CoV-2 infection is a nasopharyngeal swab. However, this method is quite uncomfortable for the patient, requires specific equipment (nose swabs and containers), and requires close proximity to health care workers, putting them at risk of infection. Developing alternative sampling strategies could decrease the burden for health care workers, help overcome potential shortages of equipment, and improve acceptability of testing by reducing patient discomfort.

**KEYWORDS** alternative testing methods, COVID-19, gargle, nasopharyngeal swab, SARS-CoV-2, spit, sputum

Severe acute respiratory syndrome coronavirus 2 (SARS-CoV-2) infection control relies on a test-trace-isolate strategy with early diagnosis and isolation of infected individuals followed by identification of contacts (1). It has led to an initial shortage of

Address correspondence to Henna Mäkelä, henna.makela@thl.fi.

Could spit be used to decrease healthcare workers exposure to SARS-CoV-2 and improve testing capacities: Findings from a Finnish exploratory study conducted by @hennamailis and @timodub at @THLorg

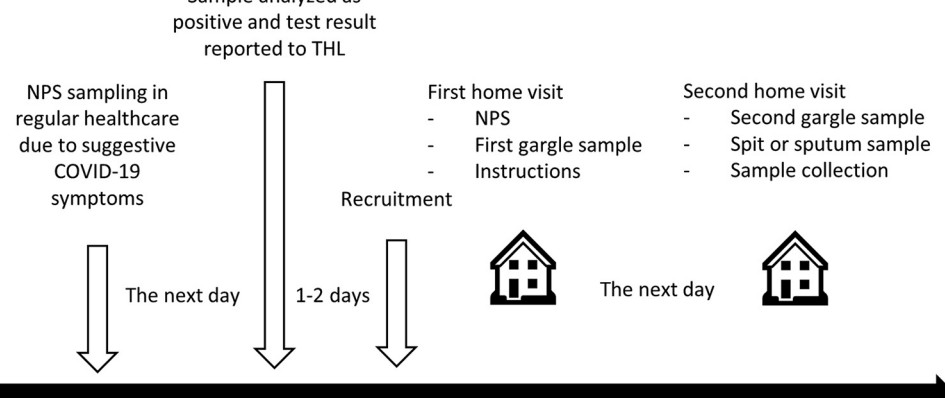

**FIG 1** Timeline of the study.

personal protective and sampling equipment, as well as increasing the workload of health care workers (HCWs) (2, 3). In Finland alone (approximately 5,500,000 inhabitants), by 20 March 2021 over 3,700,000 tests had been conducted and analyzed nationwide since the start of the coronavirus disease 2019 (COVID-19) pandemic (4, 5), including 145,000 tests in the first week of March 2021.

Collection of a nasopharyngeal swab specimen (NPS) is the gold standard for SARS-CoV-2 infection diagnosis (6). It is an unpleasant procedure, requiring close contact with a HCW and increasing the HCW's risk of infection (2, 7). The use of an alternative specimen collection method could increase testing capacities, as well as decreasing HCWs' workload and risk of infection (7). There have been several studies evaluating alternative specimen collection methods, but none of the specimen collection methods has yet superseded NPS collection (3, 8–10), even though one Finnish private health care provider now offers asymptomatic patients the possibility to self-collect a gargle specimen as an alternative to NPS collection (11). We evaluated and compared three alternative specimen collection methods that would not require close contact with a HCW and compared their sensitivity and cycle threshold ($C_T$) values with those for NPS.

## RESULTS

We enrolled 40 patients, with a mean age of 38.7 years (standard deviation [SD], 12.6 years), including 21 female patients (53%). Enrollment was performed as soon as the patients received the positive testing results, either 1 day ($n$ = 27/40 [67.5%]) or 2 days ($n$ = 13/40 [32.5%]) after they had been sampled by NPS for COVID-19 diagnosis (Fig. 1). Thirty-one patients had been symptomatic since disease onset (see Table S1 in the supplemental material), with most prevalent symptoms being fatigue (86%), headache (81%), and cough (79%). At the time of specimen collection, only 24 patients were symptomatic, with the most prevalent symptoms being cough (44%), anosmia (42%), and headache (40%).

We collected 40 concomitant NPS and gargle specimens on the recruitment day, as well as 40 next-morning gargle specimens. Of 22 patients assigned to the spit specimen collection group, all of them gave back specimens; of 18 patients assigned to the sputum group, 2 patients could not produce sputum, and thus only 16 sputum samples were included in the analysis.

All specimens were generally as sensitive as NPS to diagnose SARS-CoV-2 infection. The morning spit specimens showed the highest sensitivity (sensitivity, 1.00 [95% confidence interval [CI], 1.00 to 1.00]), followed by gargle specimens, regardless of when they had been collected (0.97 [95% CI, 0.92 to 1,00]). The sputum specimens had the lowest sensitivity (0.94 [95% CI, 0.87 to 1.00]) (Table 1).

**TABLE 1** Sensitivity and specificity of tested sampling methods

| Specimen type and result | No. with NPS result of: | | Sensitivity (95% CI) | Specificity (95% CI) | Corrected sensitivity (95% CI) | Corrected specificity (95% CI) | Cohen's kappa (95% CI) | AUC (95% CI) |
|---|---|---|---|---|---|---|---|---|
| | Positive | Negative | | | | | | |
| Gargle 1 | | | | | | | | |
| Positive | 37 | 1 | 0.97 (0.92–1.00) | 0.50 (0.00–1.00) | 0.97 (0.02–1.00) | 0.50 (0.00–1.00) | 0.47 (−0.15–1.00) | 0.74 (0.47–1.00) |
| Negative | 1 | 1 | | | | | | |
| Gargle 2 | | | | | | | | |
| Positive | 37 | 1 | 0.97 (0.92–1.00) | 0.50 (0.00–1.00) | 1.00 (1.00–1.00) | 1.00 (0.87–1.00) | 0.47 (−0.15–1.00) | 0.74 (0.47–1.00) |
| Negative | 1 | 1 | | | | | | |
| Sputum | | | | | | | | |
| Positive | 13 | 1 | 0.94 (0.83–1.00) | 0 | | | −0.09 (−0.22–0.04) | 0.54 (0.50–0.61) |
| Negative | 2 | 0 | | | | | | |
| Spit | | | | | | | | |
| Positive | 21 | 0 | 1.00 (1.00–1.00) | 1.00 (1.00–1.00) | | | 1.00 (1.00–1.00) | 1 (1–1) |
| Negative | 0 | 1 | | | | | | |

**TABLE 2** Comparison of $C_T$ values obtained with alternative samples versus NPS

| Sample type | No. of samples | $C_T$ (mean $\pm$ SD) | $P^a$ |
|---|---|---|---|
| NPS | 40 | 22.07 $\pm$ 4.63 | Reference |
| Gargle 1 | 40 | 29.89 $\pm$ 4.34 | <0.001 |
| Gargle 2 | 40 | 29.25 $\pm$ 3.99 | <0.001 |
| Sputum | 16 | 25.82 $\pm$ 9.21 | 0.28 |
| Spit | 22 | 23.51 $\pm$ 4.57 | 0.11 |

[a]Paired $t$ test.

We compared $C_T$ values obtained from alternative specimens to NPS (Table 2). NPS had the lowest $C_T$ value (22.07; SD, 4.63), although it was not significantly lower than values for sputum (25.82 [SD, 9.21]; $P = 0.28$) and spit (23.51 [SD, 4.57]; $P = 0.11$) specimens. Both gargle specimens had statistically significantly higher $C_T$ values, compared to NPS (Table 2). All patients' different specimen results are presented in Fig. 2.

## DISCUSSION

We evaluated self-collection of alternative specimens, i.e., gargle, spit, and sputum specimens, and compared their sensitivity and $C_T$ values to those for NPS for SARS-CoV-2 infection diagnosis. All specimen collection methods were comparably sensitive as NPS, with sensitivities exceeding 90%. Compared to NPS, gargle specimens had significantly higher $C_T$ values, likely due to dilution by gargling water, while spit and sputum specimens collected on the following day had higher, although not significantly different, $C_T$ values. Additionally, sputum samples appeared more challenging to collect among patients with milder symptoms. Therefore, spit would be the most suitable alternative specimen to NPS for SARS-CoV-2 infection diagnosis.

Since the beginning of the COVID-19 pandemic, several studies or meta-analyses have investigated the potential use of saliva specimens for SARS-CoV-2 infection diagnosis

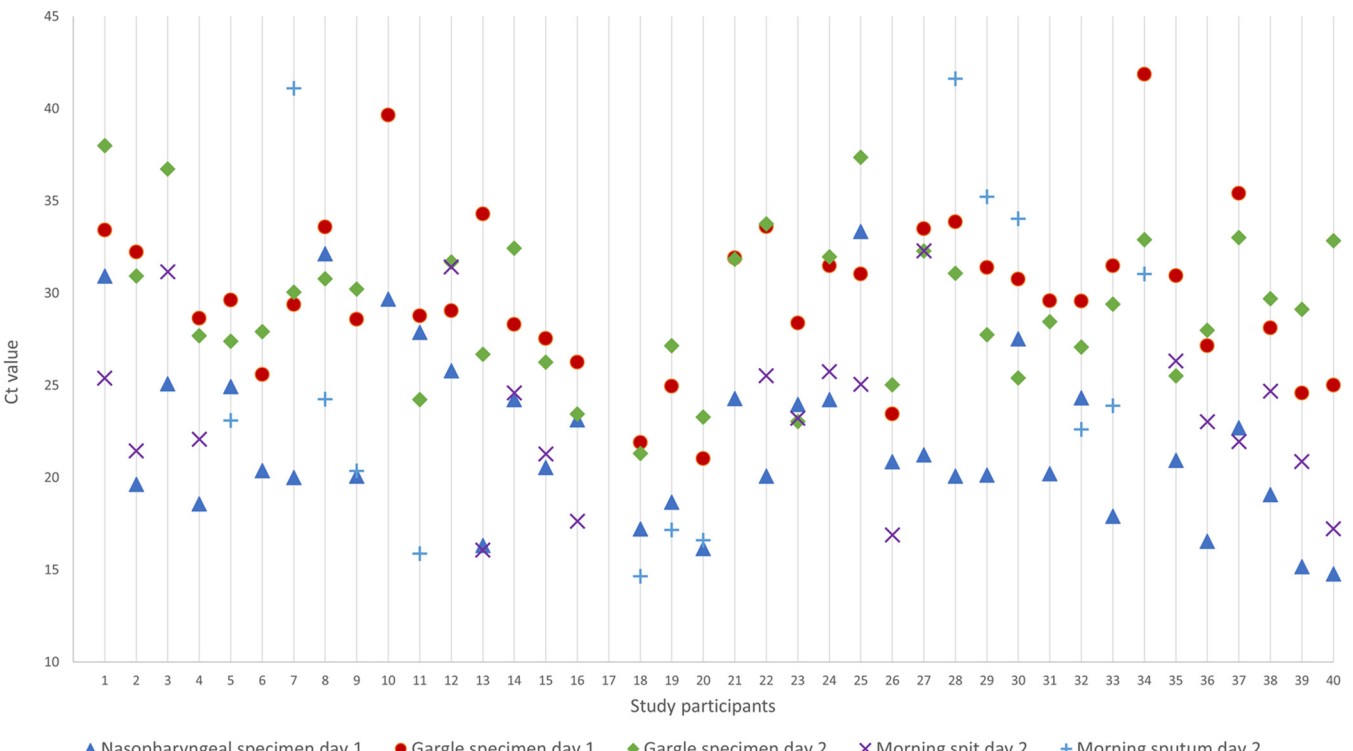

**FIG 2** $C_T$ values depending on specimen type and time of collection.

among both symptomatic (2, 7, 8, 12–17) and asymptomatic (2, 7, 8, 12–14, 17, 18) patients. The European Centre for Disease Prevention and Control (ECDC) and the Centers for Disease Control and Prevention (CDC) have approved the use of saliva or sputum samples as diagnostic specimens for COVID-19 testing for patients with a productive cough (19, 20).

There have been mixed results, considering differences in $C_T$ values between NPS and saliva specimens (8, 15, 16, 18). These inconclusive results can be caused by various factors; morning saliva samples might have higher viral loads, compared to those from the rest of the day (14), and levels of SARS-CoV-2 in saliva samples correlate with COVID-19 symptom severity (21). Additionally, the early-morning posterior oropharyngeal spitting technique has been considered to have higher sensitivity for SARS-CoV-2 infection diagnosis, compared to NPS (7).

Previously, gargle specimens were also estimated to be sensitive for SARS-CoV-2 infection diagnosis, compared to NPS, including 50 inpatients (22) and outpatients (3, 23) with confirmed COVID-19. $C_T$ values for gargle specimens were higher than those for NPS in those three studies, including one with 19,620 outpatients that is in line with findings in our study (3, 22, 23). Interestingly, Goldfarb et al., when analyzing 40 COVID-19 outpatients, concluded that gargle specimens had higher sensitivity than saliva specimens (97.5% [95% CI, 86.8 to 99.9%], compared to 78.8% [95% CI, 61.0 to 91.0%]), which is inconsistent with our results. One possible explanation for this could be that patients were instructed to gargle for a longer time in this study (3). Levican-Asenjo et al. and Malcynski et al. showed that diagnostic results and $C_T$ values were comparable between sputum specimen and NPS tests during the first 10 days after COVID-19 diagnosis (24, 25). Our results were also consistent with that study.

The main benefit of alternative specimen collection would be to avoid close contact with a HCW and to avoid an unpleasant procedure. Overall, it would also increase the willingness to apply for SARS-CoV-2 testing and the allocation of current resources for HCWs. However, other practicalities must be considered in developing an alternative sample collection method. Morning spit collection before any food or water intake appears to show the best performance but could not be offered in all situations (appointment for sampling at a later time of day, for example). However, offering this alternative to self-quarantine patients feeling unwell could be a suitable use of the morning spit collection method, assuming that the risk of slightly delayed contact tracing did not outweigh the decreased risk of exposure for HCWs.

The main limitations of our work were small sample size (40 participants in total) and the fact that we did not analyze whether delayed transport or extended storage before analysis could possibly hamper sensitivity; all samples were transported and processed on the same day. Furthermore, the volume and timing of samples (e.g., the second gargle sample after fluid intake) had some variation, which could have an effect on the analysis. Additionally, because the study sample consisted of participants with recent positive results, most patients still tested positive with most methods, meaning that the values obtained regarding specificity should be interpreted with caution. However, we conducted this study as an exploratory assessment of alternative specimen collection methods and focused on patients with the most common clinical picture of COVID-19, with mild symptoms, because they are the ones who would most benefit from noninvasive alternative specimen collection. In addition, we managed to collect all samples within 1 to 2 days after diagnosis, while patients were still in the acute phase of the disease.

## CONCLUSION

Among gargle, spit, and sputum specimens, morning collection of spit before any food or water intake or teeth brushing appeared to be the most suitable alternative specimen collection method, with $C_T$ values as low as those obtained with NPS and with ease of collection for patients with mild symptoms. Our findings are promising but warrant further investigations with larger study populations. It is important to note

that the efficiency of different RNA extraction methods can significantly vary among sample materials. For this reason, before changing sample materials, the detection method used must be carefully validated. We will consider offering to patients with mild symptoms who are seeking diagnosis in a pilot testing center the possibility of enrolling in a study assessing whether, in the general population, spit collected on the next morning has the same performance as in our exploratory sample. If so, in the long run, we could offer patients with mild symptoms the choice between at-home self-collection of spit and NPS collection at a testing center. This would not only decrease discomfort but also decrease HCWs' exposure and burden.

## MATERIALS AND METHODS

**Sample collection.** We contacted confirmed COVID-19 outpatients who had been diagnosed with SARS-CoV-2 infection by reverse transcription (RT)-PCR using NPS a few days earlier. Children <2 years of age were not eligible for participation. After calling the patients for recruitment, we visited them on the same day and the following day.

During the first home visit, we gathered informed consent, gave participants a link to an online symptom questionnaire, and collected a NPS and a gargle specimen (gargle 1). We also gave them instructions (see Appendix S1 in the supplemental material) and containers for collection of gargle (gargle 2) and, depending on the recruitment week, sputum or spit specimens the next morning. On the day following the first visit, the patients themselves collected the second gargle (gargle 2) and spit or sputum specimens. We retrieved the self-collected samples later in the day. The timeline of the study is presented in Fig. 1.

All alternative specimens were collected into a 70-ml plastic container. To collect the gargle specimens, the patients were instructed to have a sip of water and gargle it for 5 to 20 s before spitting it in the container. For spit collection, the patients had to spit continuously into the container until they filled one-half its volume; for the sputum specimens, the patients were asked to cough sputum from deep in their lungs and then spit it into the container. Both the spit and sputum specimens were to be collected in the morning before the patients ate, drank, or brushed their teeth, whereas the gargle sample was collected by the participants later in the morning. All specimens were transported at room temperature and analyzed on the same day in the Expert Microbiology Unit at the Finnish Institute for Health and Welfare (THL). The Finnish communicable diseases law and the law on the duties of the THL allowed the implementation of this noninvasive research without seeking further ethical approval (26).

**Laboratory methods.** RNA extraction from samples was performed using the Chemagic Viral300 DNA/RNA kit H96 (PerkinElmer) according to the manufacturer's instructions. A sample volume of 300 $\mu$l and an elution volume of 50 $\mu$l were used. Highly viscose gargle samples were vortex-mixed with 1 ml phosphate-buffered saline (PBS) before 300 $\mu$l was taken for RNA extraction. Real-time RT-PCR was performed using qScript XLT one-step RT-quantitative PCR (qPCR) ToughMix (Quantabio). SARS-CoV-2 was detected using the E (envelope) gene real-time RT-PCR assay. Primers and probes were based on the Corman E gene primer/probe set (27). The thermal profile for PCR was 55°C for 20 min and 95°C for 3 min, followed by 45 cycles of 95°C for 15 s and 58°C for 1 min, using a CFX thermal cycler (Bio-Rad).

**Statistical analysis.** We used NPS RT-PCR test results as the reference and $C_T$ values as surrogates for viral load analysis. We used paired $t$ tests to compare the measured $C_T$ values between NPS and gargle specimens and between NPS and sputum/spit specimens. Standard methods were used to calculate sensitivity and specificity of the other diagnostic tests (index tests) from saliva and sputum/spit specimens. We used McNemar's exact test to assess the differences in sensitivity and specificity.

We calculated Cohen's kappa coefficient to evaluate the agreement between the reference and the index tests. The area under the receiver operating characteristic curve (AUC) and 95% CI were reported. Because of the imperfect reference test, latent class analysis was used as a correction method to adjust the estimated sensitivity and specificity of the index tests based on the existing sensitivity and specificity of the reference test. Model selection was based on the Bayesian information criterion (BIC). Data analysis was performed using R software (version. 3.6.0).

## SUPPLEMENTAL MATERIAL

Supplemental material is available online only.
**SUPPLEMENTAL FILE 1**, PDF file, 0.4 MB.

## ACKNOWLEDGMENTS

We thank Helsinki city and the Helsinki Epidemiological Unit for sharing data on the COVID-19 patients. We are also grateful to the THL virology laboratory staff members who analyzed the specimens among other work.

We declare no conflicts of interest.

Author contributions were as follows: study design, H.M., L.H., T.V., H.N., N.I., O.H., C.S.-K., and T.D.; statistical analysis, T.V.; specimen collection and logistics, E.P., H.M.,

L.H., and T.D.; laboratory analysis, N.I., K.L., and the THL virology laboratory; writing and editing, all authors.

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
