## [Reviewer comments · Microbiology Spectrum]

Microbiology Spectrum

Detection of SARS-CoV-2 infection in gargle, spit and sputum specimens

Eero Poukka, Henna Mäkelä, Lotta Hagberg, Thuan Vo, Hanna Nohynek, Niina Ikonen, Kirsi Liitsola, Otto Helve, Carita Savolainen-Kopra, and Timothée Dub

Corresponding Author(s): Henna Mäkelä, Finnish institute for Health and Welfare

Review Timeline:

Submission Date:	April 15, 2021
Editorial Decision:	May 14, 2021
Revision Received:	July 27, 2021
Accepted:	July 27, 2021

Editor: Samuel Campos

Reviewer(s): The reviewers have opted to remain anonymous.

Transaction Report:

DOI: <https://doi.org/10.1128/Spectrum.00035-21>

May 14, 2021

Dr. Henna Mäkelä
Finnish institute for Health and Welfare
Mannerheimintie 166
Helsinki
Finland

Re: Spectrum00035-21 (Detection of SARS-CoV-2 infection in gargle, spit and sputum specimens)

Dear Dr. Henna Mäkelä:

Thank you for submitting your manuscript to Microbiology Spectrum. As you will see the reviewers support publication of a revised paper. Please revise the paper along the lines suggested by the reviewers.

Although one major concern raised by Reviewer #2 was the relatively small sample size, we realize that inclusion of additional samples may not be possible in these times. Rather, the greater concern would be thorough analytical validation of the alternative matrices, a.e. gargle, saliva and sputum compared to UTM, as outlined by reviewer #2.

When submitting the revised version of your paper, please provide (1) point-by-point responses to the issues raised by the reviewers as file type "Response to Reviewers," not in your cover letter, and (2) a PDF file that indicates the changes from the original submission (by highlighting or underlining the changes) as file type "Marked Up Manuscript - For Review Only". Please use this link to submit your revised manuscript - we strongly recommend that you submit your paper within the next 60 days or reach out to me for further time. Detailed information on submitting your revised paper are below.

Link Not Available

Sincerely,

Samuel Campos

Journals Department
Reviewer comments:

Reviewer #1 (Comments for the Author):

Comparative studies of nasopharyngeal specimens with different alternative collection methods to detect SARS-CoV-2 are still rare. That is why, in times of limited diagnostic resources and evident risks for health care workers during sample collection, this study provides a valuable contribution to overcome the challenges in SARS-CoV-2 diagnostics.

There are minor orthographic and grammatical mistakes in the manuscript.

Line 36: In the second sentence, the authors refer to the virus SARS-CoV-2, not to the disease.

Line 42: Abbreviation "HCW" is missing.

Line 51-54: There are also English publications/reports from diagnostic laboratories, which had to switch early in the pandemic from NPS to gargle specimen because of the lack of swabs.

Line 64: The exact collection of NPS is not described, e.g. which swabs have been used.

Line 67/Figure 1: The timeline in Fig. 1 suggests that the second gargle/spit or sputum sample and the sample collection by the authors have been performed on the same day. In the manuscript it is said, that the self-collected specimens were collected on the next day.

Line 69-71: A specified volume of gargling fluid would be better for evaluation. Nevertheless, this point should be mentioned as a limitation.

Line 73-75: At which point has the gargle specimen collection been performed? If at some other point than the spit and sputum specimens (after food intake?), might there be some factors influencing the PCR performance of gargle samples? This could be seen by variances of the internal control, if used, or even invalid results.

Line 87: Please correct primer instead of premier.

Line 92/Table 2: As solely the paired t-test is specified in the table, the Wilcoxon signed-rank test should be removed. Table 1: The authors performed many reasonable statistics (Table 1.), which could be interpreted more thoroughly. E.g. the specificity and Cohen's kappa for sputum seems to be biased by the study cohort.

Line 129-131: Would prefer to conclude "all specimen collection methods were comparably sensitive as NPS" and not as sensitive as NPS.

Line 150-154: Golfard et al. describes a collection time of in total 30 sec for gargle sample collection, which is longer than mentioned in line 70 of this manuscript. Moreover, this information is missing in the instruction for sampling in attachment 1. and could be one possible explanation for discrepancies between sputum and gargle specimens compared to Golfard.

Line 157: Check the specification of the population, some information appears to be missing.

Conclusion: Spit collection is concluded as the most suitable collection method, assuming the collection is performed before any food and water intake or teeth brushing. This additional assumption might be a limitation as well. On the one hand, people coming to a test center might have already performed any of the before mentioned. On the other hand, in case of an at home self-collection method, as mentioned in line 181, the question arises how and when the people get the containers and the sampling instruction to ensure timely diagnostics. The pros and cons of the different collection methods depending on different collection settings worthy of discussion.

Reviewer #2 (Comments for the Author):

Poukka et al. present clinical validation data comparing SARS-CoV-2 qPCR results for different matrices to the gold standard, nasopharyngeal swabs. The dataset comprises of 40 patients, with specimens concomitantly collected, though for sputum and saliva the cohort was split in two. The data is presented clearly, illustration should be optimized.

Comparison data for different specimen-types, such as presented in this study, is highly useful for diagnostic practice, especially considering the current shortages of swab collection kits and screening programs that entail frequent (at least mildly invasive) sample collection. Alternative sampling methods help conserve resources and compliance.

I want to point out the following issues:

- Grammar errors need to be corrected and overall writing style improved.

- The overall size of the cohort is small (40 patients), and even smaller for comparisons including sputum and saliva. The authors already mention this as a limitation. A larger cohort would have been beneficial for the overall value of the data.

- Matrix and assay validation: As far as can be deduced from the manuscript, the authors use a lab-developed qPCR test based on Corman et al.'s E-Sarbeco assay. They provide no validation data for the particular workflow (i.e. Perkin elmer chemagic extraction, followed by amplification by Quantabio one-step master, followed by detection on a Biorad CFX) used in this study. The semi-quantitative comparison of measurements further complicates this matter.

As a precondition to be able to analyze these results, I expect: Analytic evaluation of detection limits in UTM and in all other matrices used (i.e. how does the matrix itself influence detection of a known amount of viral RNA). Evaluation of quantitative results and defining a linear range and limit of quantification for all matrices (i.e. within what range can ct-values be compared with each other and how will the matrix influence PCR efficiency and thereby bias quantitative comparison).

- When using alternative matrices such as gargling solution, saliva and sputum, problems may occur during nucleic acid extraction depending on the method used. In our own experience, especially with gargling solution we found that many nucleic acid extraction methods require additional pre-treatment of samples to avoid a considerable loss of performance. This ranges from adding salts to pre-treating with fairly strong lysis buffers. In this particular case, the Perkin Elmar product seems to perform quite well without pre-treatment. For any users who might want to replicate the method with different setups, I would recommend stressing in the manuscript that any non-IVD certified workflow needs to be rigorously validated before implementation in diagnostics. One can not simply swap out UTM for gargle/saliva.

- Figure 2-5 should be combined to a single figure, either in panels or as a single graph. The current illustration of the data looks unrefined.
- I suggest shortening 'introduction', 'discussion' and 'conclusion'. E.g. there is no need to introduce COVID-19 in every paper.
- I suggest using the word "saliva" instead of "spit", when talking about the material, not the process.
- Line 131: It should not be suggested that lower viral loads in resp. materials are indicative of mild disease or vice versa.
- For clarification: were participants really asked to produce 30 mL of saliva (Lines 73 and 173)? Why did you require quite that much? I strongly doubt that patients will prefer this procedure over an NP swab, especially when done repeatedly for screening.

Staff Comments:

Preparing Revision Guidelines

For complete guidelines on revision requirements, please see the Instructions to Authors at [link to page]. **Submissions of a paper that does not conform to Microbiology Spectrum guidelines will delay acceptance of your manuscript.**

Please return the manuscript within 60 days; if you cannot complete the modification within this time period, please contact me. If you do not wish to modify the manuscript and prefer to submit it to another journal, please notify me of your decision immediately so that the manuscript may be formally withdrawn from consideration by Microbiology Spectrum.

If you would like to submit an image for consideration as the Featured Image for an issue, please contact Spectrum staff.

If your manuscript is accepted for publication, you will be contacted separately about payment when the proofs are issued; please follow the instructions in that e-mail. Arrangements for payment

must be made before your article is published. For a complete list of **Publication Fees**, including supplemental material costs, please visit our website.

Review

Detection of SARS-CoV-2 infection in gargle, spit and sputum specimens

Eero Poukka^{1*}, Henna Mäkelä^{1*}, Lotta Hagberg², Thuan Vo^{1,3}, Hanna Nohynek¹, Niina Ikonen^{2,3}, Kirsi Liitsola², Otto Helve¹, Carita Savolainen-Kopra², Timothée Dub¹

Comparative studies of nasopharyngeal specimens with different alternative collection methods to detect SARS-CoV-2 are still rare. That is why, in times of limited diagnostic resources and evident risks for health care workers during sample collection, this study provides a valuable contribution to overcome the challenges in SARS-CoV-2 diagnostics.

There are minor orthographic and grammatical mistakes in the manuscript.

Line 36: In the second sentence, the authors refer to the virus SARS-CoV-2, not to the disease.

Line 42: Abbreviation "HCW" is missing.

Line 51-54: There are also English publications/reports from diagnostic laboratories, which had to switch early in the pandemic from NPS to gargle specimen because of the lack of swabs.

Line 64: The exact collection of NPS is not described, e.g. which swabs have been used.

Line 67/Figure 1: The timeline in Fig. 1 suggests that the second gargle/spit or sputum sample and the sample collection by the authors have been performed on the same day. In the manuscript it is said, that the self-collected specimens were collected on the next day.

Line 69-71: A specified volume of gargling fluid would be better for evaluation. Nevertheless, this point should be mentioned as a limitation.

Line 73-75: At which point has the gargle specimen collection been performed? If at some other point than the spit and sputum specimens (after food intake?), might there be some factors influencing the PCR performance of gargle samples? This could be seen by variances of the internal control, if used, or even invalid results.

Line 87: Please correct primer instead of premier.

Line 92/Table 2: As solely the paired t-test is specified in the table, the Wilcoxon signed-rank test should be removed.

Table 1: The authors performed many reasonable statistics (Table 1.), which could be interpreted more thoroughly. E.g. the specificity and Cohens's kappa for sputum seems to be biased by the study cohort.

Line 129-131: Would prefer to conclude “all specimen collection methods were comparably sensitive as NPS” and not as sensitive as NPS.

Line 150-154: Golfard et al. describes a collection time of in total 30 sec for gargle sample collection, which is longer than mentioned in line 70 of this manuscript. Moreover, this information is missing in the instruction for sampling in attachment 1. and could be one possible explanation for discrepancies between sputum and gargle specimens compared to Golfard.

Line 157: Check the specification of the population, some information appears to be missing.

Conclusion: Spit collection is concluded as the most suitable collection method, assuming the collection is performed before any food and water intake or teeth brushing. This additional assumption might be a limitation as well. On the one hand, people coming to a test center might have already performed any of the before mentioned. On the other hand, in case of an at home self-collection method, as mentioned in line 181, the question arises how and when the people get the containers and the sampling instruction to ensure timely diagnostics. The pros and cons of the different collection methods depending on different collection settings worthy of discussion.

Dear editor of Microbiology Spectrum,

Thank you for your and reviewers' comments. They were of great help to improve the manuscript.

We have considered the comments and edited the manuscript accordingly. Please find our point by point response below:

Reviewer #1 (Comments for the Author):

Comparative studies of nasopharyngeal specimens with different alternative collection methods to detect SARS-CoV-2 are still rare. That is why, in times of limited diagnostic resources and evident risks for health care workers during sample collection, this study provides a valuable contribution to overcome the challenges in SARS-CoV-2 diagnostics.

There are minor orthographic and grammatical mistakes in the manuscript.

- We checked the grammatical mistakes in the manuscript and made corrections when appropriate.

Line 36: In the second sentence, the authors refer to the virus SARS-CoV-2, not to the disease.

- The first paragraph was deleted as suggested by reviewer #2

Line 42: Abbreviation "HCW" is missing.

- Corrected

Line 51-54: There are also English publications/reports from diagnostic laboratories, which had to switch early in the pandemic from NPS to gargle specimen because of the lack of swabs.

- Our search on Pubmed and Google scholar failed to identify such reports. Is it possible that they were removed?

Line 64: The exact collection of NPS is not described, e.g. which swabs have been used.

- Thanks for the comment. We used the diagnostic swabs that were available for us. Unfortunately, we are unable to provide this information, as supply of the National Health Institute was limited at the time we conducted this work.

Line 67/Figure 1: The timeline in Fig. 1 suggests that the second gargle/spit or sputum sample and the sample collection by the authors have been performed on the same day. In the manuscript it is said, that the self-collected specimens were collected on the next day.

- The second gargle/spit sample and the sample collection were done on the same day. We made some changes to this section and hope it is now clearer.

Line 69-71: A specified volume of gargling fluid would be better for evaluation. Nevertheless, this point should be mentioned as a limitation.

- This was added as a limitation

Line 73-75: At which point has the gargle specimen collection been performed? If at some other point than the spit and sputum specimens (after food intake?), might there be some factors influencing the PCR performance of gargle samples? This could be seen by variances of the internal control, if used, or even invalid results.

- Thanks for this insightful comment! We clarified this section and added timing of sampling as a limitation

Line 87: Please correct primer instead of premier.

- Corrected

Line 92/Table 2: As solely the paired t-test is specified in the table, the Wilcoxon signed-rank test should be removed. Table 1: The authors performed many reasonable statistics (Table 1.), which could be interpreted more thoroughly. E.g. the specificity and Cohens's kappa for sputum seems to be biased by the study cohort.

- We thank the reviewer for noticing that we had forgotten to remove the Wilcoxon signed-rank-test. This is now done.

Regarding the fact that specificity must be interpreted in light of the study sample, it is a very good point too. It is now mentioned in the discussion as a limitation.

Line 166: Additionally, as the study sample consisted of participants with a recent positive result, most patients were still positive with most methods, meaning that the values obtained regarding specificity should be interpreted with caution.

Line 129-131: Would prefer to conclude "all specimen collection methods were comparably sensitive as NPS" and not as sensitive as NPS.

- The change was made

Line 150-154: Golfard et al. describes a collection time of in total 30 sec for gargle sample collection, which is longer than mentioned in line 70 of this manuscript. Moreover, this information is missing in the instruction for sampling in attachment 1. and could be one possible explanation for discrepancies between sputum and gargle specimens compared to Golfard.

- Thanks for the valuable comment! This is now mentioned in the discussion.

Line 157: Check the specification of the population, some information appears to be missing.

We thank the reviewer for this valuable comment.

Levican et al included some outpatients and Malcynski et al do not specify the N of hospitalized patients, only samples (50 samples). The section was corrected.

Conclusion: Spit collection is concluded as the most suitable collection method, assuming the collection is performed before any food and water intake or teeth brushing. This additional assumption might be a limitation as well. On the one hand, people coming to a test center might have already performed any of the before mentioned. On the other hand, in case of an at home self-collection method, as mentioned in line 181, the question arises how and when the people get the containers and the sampling instruction to ensure timely diagnostics. The pros and cons of the different collection methods depending on different collection settings worthy of discussion.

- *We thank the reviewer for bringing up this very relevant point. We have now added a paragraph to the discussion section discussing the pros and cons of alternate sampling collection, starting from line 164: "However, other practicalities must be considered in developing an alternative sample collection method offer. Morning spit collection before any food or water intake appears to be showing the best performance but could not be offered in all situations (appointment for sampling at a later time of the day, for example). However, offering this alternative in self-quarantine patients feeling unwell could be a suitable use of morning spit collection method, assuming the risk of slightly delayed contact-tracing does not outweigh the decreased risk of exposure for HCW."*

Reviewer #2 (Comments for the Author):

Poukka et al. present clinical validation data comparing SARS-CoV-2 qPCR results for different matrices to the gold standard, nasopharyngeal swabs. The dataset comprises of 40 patients, with specimens concomitantly collected, though for sputum and saliva the cohort was split in two. The data is presented clearly, illustration should be optimized.

Comparison data for different specimen-types, such as presented in this study, is highly useful for diagnostic practice, especially considering the current shortages of swab collection kits and screening programs that entail frequent (at least mildly invasive) sample collection. Alternative sampling methods help conserve resources and compliance.

I want to point out the following issues:

- Grammar errors need to be corrected and overall writing style improved.

- We thank the reviewer for his valuable comments. We went through the manuscript and proceeded to grammatical and styling improvement.

- The overall size of the cohort is small (40 patients), and even smaller for comparisons including sputum and saliva. The authors already mention this as a limitation. A larger cohort would have been beneficial for the overall value of the data.

- The reviewer #2 has a valid point but unfortunately we cannot present larger study population at the moment

- Matrix and assay validation: As far as can be deduced from the manuscript, the authors use a lab-developed qPCR test based on Corman et al.'s E-Sarbeco assay. They provide no validation data for the

particular workflow (i.e. Perkin elmer chemagic extraction, followed by amplification by Quantabio one-step master, followed by detection on a Biorad CFX) used in this study. The semi-quantitative comparison of measurements further complicates this matter. As a precondition to be able to analyze these results, I expect: Analytic evaluation of detection limits in UTM and in all other matrices used (i.e. how does the matrix itself influence detection of a known amount of viral RNA). Evaluation of quantitative results and defining a linear range and limit of quantification for all matrices (i.e. within what range can ct-values be compared with each other and how will the matrix influence PCR efficiency and thereby bias quantitative comparison).

- When using alternative matrices such as gargling solution, saliva and sputum, problems may occur during nucleic acid extraction depending on the method used. In our own experience, especially with gargling solution we found that many nucleic acid extraction methods require additional pre-treatment of samples to avoid a considerable loss of performance. This ranges from adding salts to pre-treating with fairly strong lysis buffers. In this particular case, the Perkin Elmar product seems to perform quite well without pre-treatment. For any users who might want to replicate the method with different setups, I would recommend stressing in the manuscript that any non-IVD certified workflow needs to be rigorously validated before implementation in diagnostics. One can not simply swap out UTM for gargle/saliva.

- To demonstrate the performance of our in-house SARS-CoV-2 PCR test, we have successfully participated in several international external quality assessment programs for SARS-CoV-2.

Unfortunately, no samples with known viral load were available for more detailed analyses of the influence of matrix itself and evaluation of the detection limits. However, we think that our results showed that gargle, sputum and spit can be successfully used as an alternative material for SARS-CoV-2 PCR testing in the method we used.

We agree that matrix used can effect on the sensitivity of the assay, especially on the efficiency of RNA extraction and validation is needed before changing the matrix. We added in the conclusion (line 189): It is important to note that the efficiency of different RNA extraction methods can significantly vary between sample materials. For this reason, before changing sample material the detection method used must be carefully validated.

- **Figure 2-5 should be combined to a single figure, either in panels or as a single graph. The current illustration of the data looks unrefined.**

- *We thank the reviewer for this suggestion. We are now displaying results in one single figure.*

- **I suggest shortening 'introduction', 'discussion' and 'conclusion'. E.g. there is no need to introduce COVID-19 in every paper.**

- *We proceeded to several cuts through out the manuscript following the reviewer's suggestion.*

- **I suggest using the word "saliva" instead of "spit", when talking about the material, not the process.**

- *We would rather refer to a spit sample instead of saliva in order to make sure that reader understand that these samples were produced actively by the patient followed by the action of spitting. It helps differentiate from samples that could have been collected using a salivette or other tools. This is a decision we took during study design as with other colleagues, we needed to make sure we had a consistent definition of what samples were.*

- **Line 131: It should not be suggested that lower viral loads in resp. materials are indicative of mild disease or vice versa.**

- *The comment was valuable and we removed the sentence from the manuscript*

- **For clarification: were participants really asked to produce 30 mL of saliva (Lines 73 and 173)? Why did you require quite that much? I strongly doubt that patients will prefer this procedure over an NP swab, especially when done repeatedly for screening.**

- *Thank you for the valid point. We collected 30 ml of spit to test different laboratory methods if needed. For diagnostic purposes 1 ml is usually enough.*

July 27, 2021

Dr. Henna Mäkelä
Finnish institute for Health and Welfare
Mannerheimintie 166
Helsinki
Finland

Re: Spectrum00035-21R1 (Detection of SARS-CoV-2 infection in gargle, spit and sputum specimens)

Dear Dr. Henna Mäkelä:

Thank you for submitting your revised manuscript to Microbiology Spectrum, I am pleased to inform you that your manuscript has been accepted and I am forwarding it to the ASM Journals Department for publication. You will be notified when your proofs are ready to be viewed.

Sincerely,

Samuel Campos
Editor, Microbiology Spectrum
